# Characterization and prediction of non-melanoma skin cancer incidence in China: Joinpoint regression and age-period-cohort model

**Su Liang[1†], Tao Sun[2], Xue Song Jia[1], Juan Mei Cao◯[1]\*, Xue Wang[1]\***

**1** Department of Dermatology, The First Affiliated Hospital of Shihezi University, Shihezi, Xinjiang, China, **2** Department of STD and AIDS Prevention, Shihezi Center for Disease Control and Prevention, Shihezi, Xinjiang, China

\* 124739471@qq.com (JMC), 1658436810@qq.com (XW)

## Abstract

### Objective

Understanding the non-melanoma skin cancer (NMSC) incidence and its trends in China is an important prerequisite for effective prevention and control of NMSC.

### Methods

NMSC incidence data was collected from the Annual Report of China Cancer Registry from 2005 to 2018. The Joinpoint regression model was used to estimate the average annual percent change (AAPC) and annual percent change (APC) to reflect the time trend. Age-period-cohort model with the intrinsic estimator algorithm was used to analyze age, period, and cohort effects. Bayesian age-period-cohort (BAPC) model with integrated nested laplace approximation was used for prediction.

### Results

The age-standardized incidence rate (ASIR) of NMSC increased from 1.02/ 100,000 to 1.63/100,000 from 2005 to 2018, showing an increasing trend with AAPC of 3.7% (95% CI: 2.5%, 4.9%). The ASIR was higher in men than in women, while the increase rate was reversed, and it was lower in rural than in urban areas, while AAPC was 1.15 times higher. The risk of NMSC incidence increased with age. The cohort effect was first to increase and then to decrease and the inflection point appeared in 1930-1934. The ASIR of NMSC in China will continue to rise during 2019-2035.

### Conclusion

The ASIR of NMSC in China from 2005 to 2018 showed an increasing trend with age, gender, and regional differences, and will continue to increase in the future. NMSC remains a public health problem and requires continuous attention.

†Lead author

**Data availability statement:** All relevant data are within the article and its Supporting Information files. Data can be obtained from the "Annual Report of China Cancer Registry, 2008-2021".

**Funding:** This study was supported by the National Natural Science Foundation of China (Project No: 82203956) , Corps-level Science and Technology Guiding Project (Project No: 2023ZD023) and Academy-level Science and Technology Plan of The First Affiliated Hospital of Shihezi University (Project No: QN202130). Funders play no role in the study.

**Competing interests:** The authors declare that they have no known competing financial interests or personal relationships that could have appeared to influence the work reported in this paper.

# 1. Introduction

Non-melanoma skin cancer (NMSC) is one of the most common cancers worldwide, including basal cell carcinoma and squamous cell carcinoma. NMSC not only affects quality of life for patients, but also may lead to serious complications, such as organization destruction and loss of function. According to recent reports, the incidence of NMSC ranked fifth in the world, despite the low mortality, its high incidence and treatment costs place a heavy burden on patients and healthcare systems [1]. In recent years, the incidence of NMSC has been on the rise globally, possibly due to changes in lifestyle, increased ultraviolet (UV) exposure, and an aging population. For example, UV light is considered to be a carcinogen, which can induce and promote the occurrence and development of cancer. As the most populated country in the world, the epidemiological characteristics of NMSC and its changing trends in China have drawn widespread attention. However, systematic studies on NMSC in China remain limited, which poses challenges for the development of effective public health interventions and prevention strategies.

This study obtained the data from the Annual Report of China Cancer Registry to systematically analyze the incidence of NMSC and its current situation and trend in China at different levels by Joinpoint regression, age-period-cohort and Bayesian age-period-cohort (BAPC) models. With multi-model analysis, this study will reveal the incidence characteristics of NMSC in different ages, periods and birth cohorts, explore its potential influencing factors, and predict the future trends, aiming to promote the development of effective preventive measures and strategies for NMSC.

# 2. Materials and methods

## 2.1 Data source

Data on the incidence of NMSC in China for 2005-2018 was obtained from the "Annual Report of China Cancer Registry, 2008-2021" [2–15]. The standard population data were obtained from the "Sixth National Population Census of China" published by the National Bureau of Statistics [16]. NMSC was diagnosed according to the International Classification of Diseases, 10th Revision (C44) [17]. The data were reviewed, evaluated and analyzed according to the requirements of the China Cancer Registry Guidelines and the International Agency for Research on Cancer (IARC)/International Association of Cancer Registries (IACR). The classification of urban and rural areas in this study was based on the classification criteria of the Annual Report of China Cancer Registry: urban areas were categorized as cities of prefectural level or above, and rural areas were categorized as counties and county-level cities.

## 2.2 Statistical analysis

**2.2.1 Joinpoint regression model.** Trends in NMSC were described using average annual percent change (AAPC) and annual percent change (APC) and their 95% confidence intervals (95% CI) calculated by the Joinpoint regression model. The model created segmented regressions based on the temporal characteristics of the disease distribution, divided the time into different intervals through multiple connectors, and fitted and optimized the trend within each interval to assess the characteristics of disease-specific changes within different intervals. The Monte Carlo permutation test was used to determine the number of linkage points, the location of each linkage point, and the corresponding *P* value, with a test level of α= 0.05 (two-sided test). The formula of the Joinpoint regression model is:

$$E\left[Y\middle|X\right] = e^{\beta_0 + \beta_1 + \delta(x-\tau_1)^+ + \ldots + \delta_k(x-\tau_k)}$$

In the formula, e is the natural base, $k$ represents the number of turning points, $\tau_k$ represents the unknown turning points, $\beta_0$ is the invariant parameter, $\beta_1$ is the regression coefficient, $\delta_k$ represents the regression coefficient of the kth segmentation function. When $(x - \tau_k) > 0$, $(x - \tau_1)^+ = x - \tau_k$, otherwise $(x - \tau_1)^+ = 0$.

$$\text{Calculation formula of APC}: \text{APC} = \left(e^{\beta_1} - 1\right) \times 100$$

$$\text{Calculation formula of AAPC}: AAPC = \left[exp\left(\sum \omega_i \beta_i / \sum \omega_i\right) - 1\right] \times 100$$

Where $\beta_1$ is the regression coefficient, $\omega_i$ is the width of the interval span (i.e.,the number of years included in the interval) for each partition function, $\beta_i$ is the corresponding regression coefficient for each interval.

**2.2.2 Age-period-cohort model.** To analyze the effects of age, period, and cohort on the incidence of NMSC, 5 years of age was used as one age group, and 0 ~ 89 years of age was divided into 18 age groups. The age-period-cohort model was evaluated by solving the intrinsic estimator (IE) using a Poisson log-linear model, and the model fit was comprehensively evaluated using the red pool information criterion [18]. In the model with the IE approach, age-specific incidence were appropriately re-coded into consecutive 5-year age groups (0-4, 5-9,..., 85-89), consecutive 5-year period groups for 2005-2018, and corresponding consecutive 5-year birth cohorts (1920-1924, 1925-1929,..., 2010-2014, 2015-2019).

$$Y_j = \mu + \alpha \, \text{age}_j + \beta \, \text{period}_j + \gamma \, \text{cohort}_j + \varepsilon_j$$

In which, $Y_j$ is the response variable for group j representing the net effect on NMSC incidence, $\alpha$、 $\beta$ and $\gamma$ are the coefficients for age, period, and cohort, respectively, μ is the intercept, $\varepsilon$ is the residual. To avoid expanding the birth cohort and reducing the temporal precision in describing the risk of incidence, this study used age-specific data from 2005, 2010, and 2015 for simulation of the age-period-cohort model.

**2.2.3 Bayesian age-period-cohort analysis.** The BAPC model was used to predict the NMSC incidence over the next 15 years. In the model, a second-order random walk model was used to smooth a priori age, period, and cohort effects to predict a posteriori incidence. This method incorporated the integrated nested laplace approximation (INLA), which prevented the mixing and convergence problems caused by Markov chain Monte Carlo sampling, resulting in more stable and reliable data. The details were described in the previous study [19].

Joinpoint regression analysis was performed using the Joinpoint Regression Program 4.9.1.0 software developed by the National Cancer Institute, and when the AAPC and its 95% CI were > 0, < 0, or included 0, it meant that the corresponding disease indicator showed an upward trend, a downward trend, or remained stable, respectively [20]. Age-period-cohort model was conducted using the online network analysis tool and Stata 17.0 software (Stata Corp, College Station, TX, USA). BAPC (version 0.036) and INLA (version 22.05.07) packages of R software were used for statistical predictions, and the "ggplot2" package was plotted. $P < 0.05$ was considered to be statistically significant.

## 3. Result

### 3.1 Incidence of NMSC and its trends in China, 2005–2018

From 2005 to 2018, the total number of new cases of NMSC in China was 72,572, including 37,299 in men (51.40%), 35,273 in women (48.60%), 42,844 in urban (59.04%) and 29,250 in rural (40.96%). The crude incidence rate was 2.44/100,000, of which 2.48/100,000 for men,

2.41/100,000 for women, 2.74/100,000 for urban, and 2.07/100,000 for rural. The ASIR was 1.33/100,000, of which 1.46/100,000 for men, 1.21/100,000 for women, 1.44/100,000 for urban, and 1.15/100,000 for rural. The crude incidence rate and ASIR were higher in men and urban than in women and rural areas, respectively (Table 1).

From 2005 to 2018, the ASIR for NMSC in China increased from 1.02/100,000 to 1.63/100,000 with AAPC of 3.7% (95% CI: 2.5%, 4.9%). By sex, the ASIR in men increased from 1.18/100,000 to 1.76/100,000 with AAPC of 3.1% (95% CI: 1.7%, 4.4%), and in women increased from 0.87/100,000 to 1.55/100,000 with AAPC of 4.5% (95% CI: 3.4%, 5.6%). By region, urban ASIR increased from 1.08/100,000 to 1.70/100,000 with AAPC of 4.0% (95% CI: 2.7%, 5.3%), and rural ASIR increased from 0.85/100,000 to 1.57/100,000 with AAPC of 4.6% (95% CI: 2.8%, 6.5%) (Table 1, Table 2, Fig 1).

## 3.2 Age-period-cohort model

The effect of age on the NMSC incidence in China were consistent across the country, among men and women, and among urban and rural, although the incidence was more variable among young and middle-aged people in rural areas. In 0–-44 years, the risk of NMSC increased rapidly with age, and then slowly in 45–89 years, peaking at 85–89 years. Among those under 18 years, the risk of NMSC was higher in children aged 0–4 years. In the rural population aged 40–89 years, the age effects were consistent with the nation, but the risk fluctuated in the population aged 10–39 years, and was substantially higher in 30–34 years.

In the period effect, the national risk of NMSC gradually increased over time. The cohort effect for NMSC risk peaked in 1930-1934 with a relative risk (RR) of 1.25 (95% CI: 0.89, 1.60), declined thereafter, and then increased slightly in 1995-1999 with an RR of -1.08 (95% CI: -1.87, -0.29) (Table 3, Fig 2).

**Table 1. Incidence rate of NMSC in China from 2005 to 2018 (/100,000).**

| Years | National | | | Male | | | Female | | | Urban | | | Rural | | |
|---|---|---|---|---|---|---|---|---|---|---|---|---|---|---|---|
| | Case | Rate | ASIR | Case | Rate | ASIR | Case | Rate | ASIR | Case | Rate | ASIR | Case | Rate | ASIR |
| 2005 | 1094 | 1.99 | 1.02 | 603 | 2.17 | 1.18 | 491 | 1.81 | 0.87 | 884 | 2.17 | 1.08 | 210 | 1.47 | 0.85 |
| 2006 | 1297 | 2.18 | 1.07 | 710 | 2.37 | 1.23 | 587 | 1.99 | 0.93 | 1051 | 2.26 | 1.08 | 246 | 1.89 | 1.06 |
| 2007 | 1317 | 2.20 | 1.07 | 686 | 2.27 | 1.18 | 631 | 2.13 | 0.96 | 1084 | 2.43 | 1.14 | 233 | 1.53 | 0.83 |
| 2008 | 1688 | 2.55 | 1.18 | 910 | 2.73 | 1.35 | 778 | 2.37 | 1.01 | 1450 | 2.78 | 1.28 | 238 | 1.70 | 0.80 |
| 2009 | 1775 | 2.08 | 0.98 | 919 | 2.13 | 1.06 | 856 | 2.03 | 0.90 | 1335 | 2.32 | 1.06 | 440 | 1.57 | 0.78 |
| 2010 | 2534 | 2.03 | 1.35 | 1403 | 2.22 | 1.57 | 1131 | 1.84 | 1.15 | 1801 | 2.25 | 1.44 | 733 | 1.64 | 1.17 |
| 2011 | 3298 | 2.26 | 1.47 | 1742 | 2.37 | 1.64 | 1556 | 2.16 | 1.32 | 2238 | 2.56 | 1.58 | 582 | 1.96 | 1.29 |
| 2012 | 4570 | 2.31 | 1.50 | 2312 | 2.30 | 1.59 | 2258 | 2.31 | 1.40 | 2776 | 2.76 | 1.68 | 1794 | 1.84 | 1.28 |
| 2013 | 4982 | 2.20 | 1.39 | 2636 | 2.29 | 1.54 | 2346 | 2.10 | 1.25 | 2911 | 2.61 | 1.55 | 2071 | 1.80 | 1.22 |
| 2014 | 6740 | 2.34 | 1.45 | 3507 | 2.40 | 1.58 | 3233 | 2.28 | 1.33 | 3922 | 2.72 | 1.61 | 2818 | 1.95 | 1.26 |
| 2015 | 7800 | 2.43 | 1.48 | 3999 | 2.46 | 1.58 | 3801 | 2.40 | 1.38 | 4428 | 2.87 | 1.68 | 3372 | 2.02 | 1.28 |
| 2016 | 9518 | 2.49 | 1.49 | 4822 | 2.49 | 1.58 | 4696 | 2.50 | 1.41 | 5446 | 2.83 | 1.63 | 4072 | 2.16 | 1.35 |
| 2017 | 11280 | 2.59 | 1.54 | 5712 | 2.58 | 1.63 | 5568 | 2.59 | 1.45 | 6309 | 2.96 | 1.70 | 4971 | 2.23 | 1.38 |
| 2018 | 14,679 | 2.81 | 1.63 | 7,338 | 2.76 | 1.76 | 7,341 | 2.85 | 1.55 | 7,209 | 3.05 | 1.70 | 7,470 | 2.60 | 1.57 |
| Total | 72,572 | 2.44 | 1.33 | 37,299 | 2.48 | 1.46 | 35,273 | 2.41 | 1.21 | 42,844 | 2.74 | 1.44 | 29,250 | 2.07 | 1.15 |

ASIR, age-standardized incidence rate. NMSC, non-melanoma skin cancer.

**Table 2. Trends in NMSC incidence in China from 2005 to 2018 (%).**

| Indexes | National | Male | Female | Urban | Rural |
|---|---|---|---|---|---|
| Periods | 2005–2018 | 2005–2018 | 2005–2009 | 2005–2018 | 2005–2018 |
| APC (95%CI) | 3.7* (2.5, 4.9) | 3.1* (1.7, 4.4) | 1.5* (-4.6, 7.6) | 4.0* (2.7, 5.3) | 4.6* (2.8, 6.5) |
| t | 6.6 | 4.8 | 0.78 | 6.7 | 5.5 |
| P | < 0.001 | < 0.001 | 0.490 | < 0.001 | < 0.001 |
| Periods | | | 2009–2012 | | |
| APC (95%CI) | | | 9.8* (-19.1, 38.8) | | |
| t | | | 4.3 | | |
| P | | | 0.145 | | |
| Periods | | | 2012–2018 | | |
| APC (95%CI) | | | 3.9* (2.9, 4.8) | | |
| t | | | 11.17 | | |
| P | | | < 0.001 | | |
| AAPC (95%CI) | 3.7* (2.5, 4.9) | 3.1* (1.7, 4.4) | 4.5* (3.4, 5.6) | 4.0* (2.7, 5.3) | 4.6* (2.8, 6.5) |
| t | 6.6 | 4.8 | 8.8 | 6.7 | 5.5 |
| P | < 0.001 | < 0.001 | < 0.001 | < 0.001 | < 0.001 |

AAPC, Average Annual Percentage Change; APC, Annual Percent Change; 95%CI, 95% Confidence Interval. NMSC, non-melanoma skin cancer.

## 3.3 Bayesian age-period-cohort models

The incidence prediction of NMSC from 2019 to 2035 in China showed a gradual increase in ASIR, which was essentially the same across the country, among men and women, and urban populations.

The ASIR was predicted to rise to 4.62/100,000 (95% CI: -117.51, 126.76) in 2035, of which it will rise to 2.45/100,000 (95% CI: 0.31, 4.58) in men, 3.25/100,000 (95% CI: -13.34, 19.85) in women, 4.55/100,000 (95% CI: -54.56, 63.66) in urban, and 5.67/100,000 (95% CI: -168.69, 180.02) in rural (Fig 3).

## 4. Discussion

The aim of this study was to analyze the incidence characteristics of NMSC in China. During 2005 to 2018, the ASIR of NMSC in China showed an increasing trend nationwide, with AAPC of 3.7% (95% CI: 2.5%, 4.9%), and predictions also found that it would continue to increase by 2035. The ASIR of NMSC in men was higher than that in women, but women's increase rate was higher than men's. The ASIR was higher in urban than in rural areas, while the increase rate was opposite. These findings are valid epidemiological evidence for the prevention and control of NMSC.

The relative disadvantage in the incidence of NMSC in men and the urban population may be related to men's higher exposure to ultraviolet rays due to more outdoor activities and higher levels of industrial pollution in the city [21]. In addition, the incidence of NMSC in China has shown a significant upward trend, especially in rural areas and women. The faster increase in rural areas may be related to poor awareness of NMSC prevention, more risky sun behaviors, and insufficient sun protection measures. Ultraviolet (UV) light from sunlight is

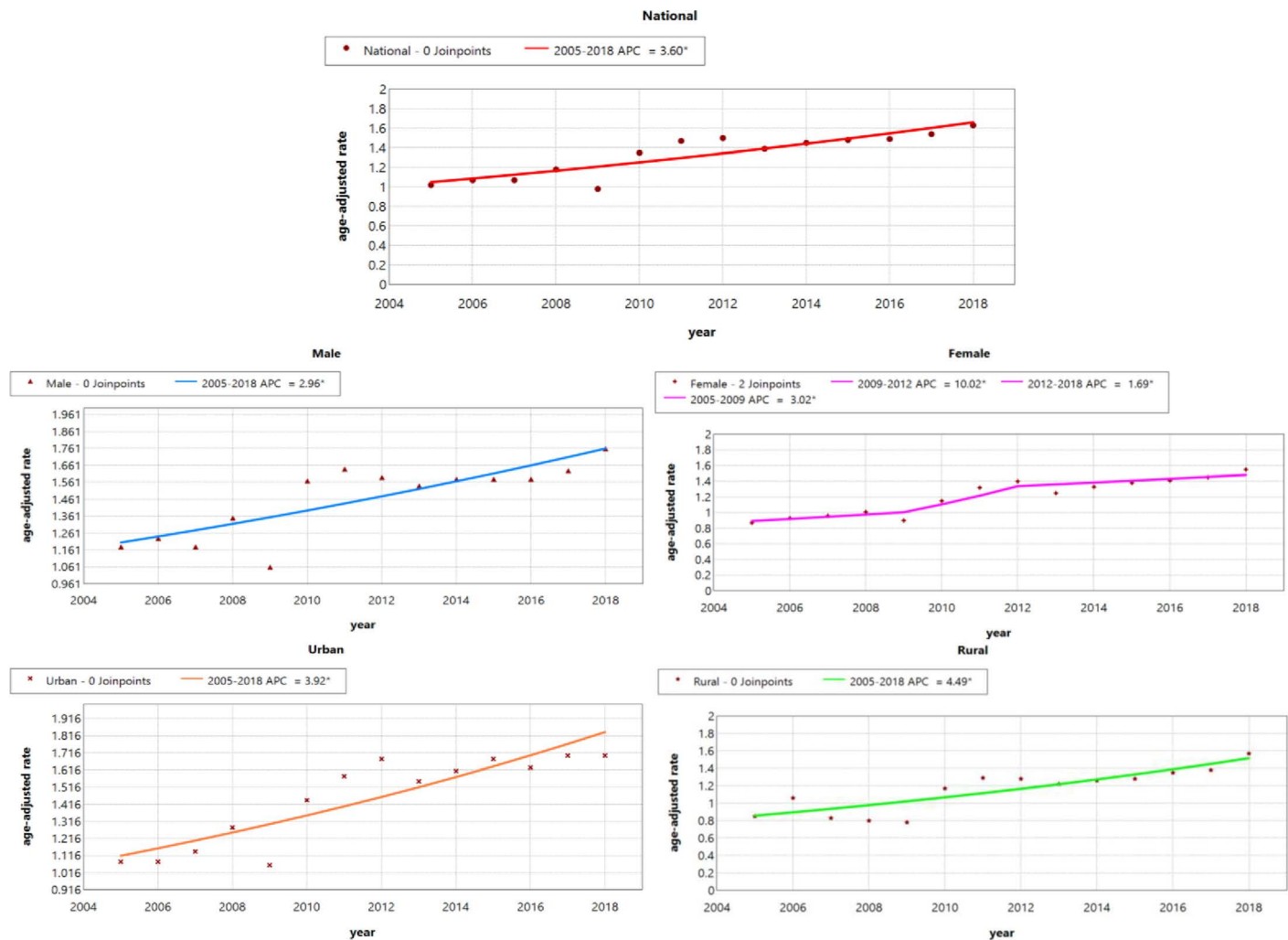

**Fig 1. Joinpoint regression in the incidence of NMSC in China, 2005–2018.** NMSC, non-melanoma skin cancer.

the major cause of NMSC development, and it is a complete carcinogen that not only initiates tumorigenesis by inducing mutations in tumor suppressor genes, but also promotes tumor progression [22]. Long-term outdoor agricultural labor in rural populations may lead to excessive UV exposure. Studies have shown that perceived cancer risk predicts preventive behaviors, and lack of perceived risk is a barrier to risk reduction. Blacks have a lower perceived risk of cancer such as breast cancer, which results in a higher cancer mortality than any other race in the U.S [23]. Patient cognitive deficiencies, such as low levels of education and lack of health awareness, increase skin cancer incidence and mortality [24]. A study found that the lower the level of education, the lower the perceived risk of skin cancer ($P = 0.0063$), the lower the concern about suffering from skin cancer ($P \leq 0.0144$), and the lower the knowledge of skin cancer prevention methods ($P \leq 0.003$) [25]. In addition, the increasing rate of NMSC incidence in women was higher than that in men, possibly related to the increase in outdoor activity in women in recent years. The lower incidence of NMSC in women than in men may be due to the fact that women are more concerned about the development of skin cancer as well as the more frequent use of sunscreen than men [26]. Therefore, it is particularly

**Table 3. NMSC incidence risk due to age, period, and cohort effects.**

| Factor | RR | 95%CI | P-value |
|---|---|---|---|
| Age (years) | | | |
| 0–4 | −1.39 | (−3.04, 0.27) | 0.101 |
| 5–9 | −2.04 | (−3.28, −0.79) | 0.001 |
| 10–14 | −1.41 | (−2.26, −0.56) | 0.001 |
| 15–19 | −1.41 | (−2.19, −0.62) | <0.001 |
| 20–24 | −1.06 | (−1.81, −0.31) | 0.006 |
| 25–29 | −0.62 | (−1.32, 0.09) | 0.085 |
| 30–34 | −0.43 | (−1.09, 0.23) | 0.205 |
| 35–39 | −0.34 | (−0.94, 0.25) | 0.259 |
| 40–44 | −0.04 | (−0.56, 0.48) | 0.878 |
| 45–49 | 0.08 | (−0.36, 0.52) | 0.724 |
| 50–54 | 0.29 | (−0.07, 0.65) | 0.114 |
| 55–59 | 0.60 | (0.32, 0.88) | <0.001 |
| 60–64 | 0.77 | (0.56, 0.98) | <0.001 |
| 65–69 | 0.89 | (0.72, 1.06) | <0.001 |
| 70–74 | 1.19 | (1.00, 1.37) | <0.001 |
| 75–79 | 1.43 | (1.18, 1.67) | <0.001 |
| 80–84 | 1.64 | (1.31, 1.98) | <0.001 |
| 85–89 | 1.84 | (1.40, 2.27) | <0.001 |
| Period | | | |
| 2005 | −1.10 | (−1.25, −0.96) | <0.001 |
| 2010 | 0.44 | (0.37, 0.50) | <0.001 |
| 2015 | 0.67 | (0.56, 0.78) | <0.001 |
| Cohort | | | |
| 1920–1924 | 1.16 | (0.38, 1.95) | 0.004 |
| 1925–1929 | 1.23 | (0.76, 1.70) | <0.001 |
| 1930–1934 | 1.25 | (0.89, 1.60) | <0.001 |
| 1935–1939 | 1.09 | (0.82, 1.35) | <0.001 |
| 1940–1944 | 0.96 | (0.76, 1.16) | <0.001 |
| 1945–1949 | 0.83 | (0.66, 1.01) | <0.001 |
| 1950–1954 | 0.76 | (0.55, 0.96) | <0.001 |
| 1955–1959 | 0.50 | (0.23, 0.77) | <0.001 |
| 1960–1964 | 0.25 | (−0.10, 0.60) | 0.159 |
| 1965–1969 | 0.10 | (−0.34, 0.53) | 0.669 |
| 1970–1974 | −0.08 | (−0.60, 0.43) | 0.750 |
| 1975–1979 | −0.31 | (−0.90, 0.29) | 0.315 |
| 1980–1984 | −0.45 | (−1.11, 0.22) | 0.188 |
| 1985–1989 | −0.79 | (−1.51, −0.07) | 0.031 |
| 1990–1994 | −0.79 | (−1.54, −0.03) | 0.041 |
| 1995–1999 | −1.08 | (−1.87, −0.29) | 0.007 |
| 2000–2004 | −0.62 | (−1.41, 0.18) | 0.129 |
| 2005–2009 | −1.05 | (−1.94, −0.16) | 0.021 |
| 2010–2014 | −1.31 | (−2.66, 0.03) | 0.056 |
| 2015–2019 | −1.65 | (−3.42, 0.12) | 0.068 |

RR, relative risk. NMSC, non-melanoma skin cancer.

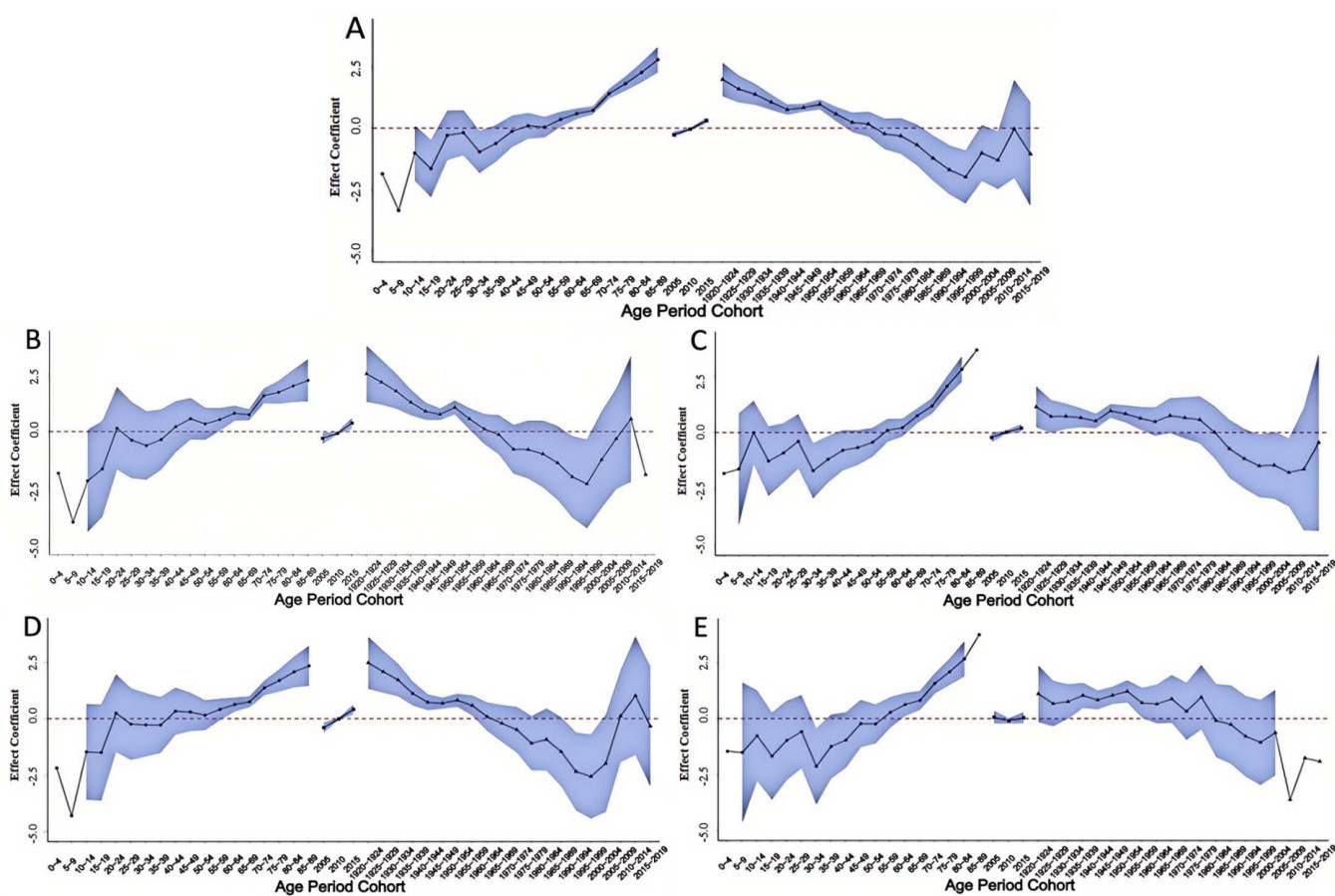

**Fig 2. Age-period-cohort model of NMSC incidence in China, 2005–2018.** (A) Nation; (B) Men; (C) Women; (D) Urban; (E) Rural. NMSC, non-melanoma skin cancer.

important to adopt differentiated prevention and intervention measures for different populations.

The effect of age on the incidence of NMSC was generally consistent across the country and among urban and rural. The risk of NMSC increased rapidly with age in the 0-44 age group, whereas it slowed down in the 45-89 years and peaked at 85-89 years of age. Especially in rural populations, the age effect was significant and highly variable, and the large increase in incidence among aged 30-34 may be related to certain external factors. As a high-risk group for occupational skin cancer, the occupational group of outdoor workers (e.g., agricultural and forestry workers, market gardeners, sailors and fishermen, construction workers and artisans, as well as road workers, pool attendants, and mountain guides) are most at risk for the development of skin cancers because they spend most of their time outdoors and are exposed to high levels of solar UV radiation. As a result of this exposure, outdoor workers are at least twice as likely as the general population to develop skin cancer [27–30]. This suggests that when developing preventive measures for NMSC, special attention should be paid to the relevant occupational groups with targeted interventions to enhance occupational protection.

In the period effect, NMSC risk increased gradually over time. This may be related to factors such as increased environmental pollution due to accelerated industrialization and intensified UV radiation. China's Environmental Performance Index (EPI) score decreased

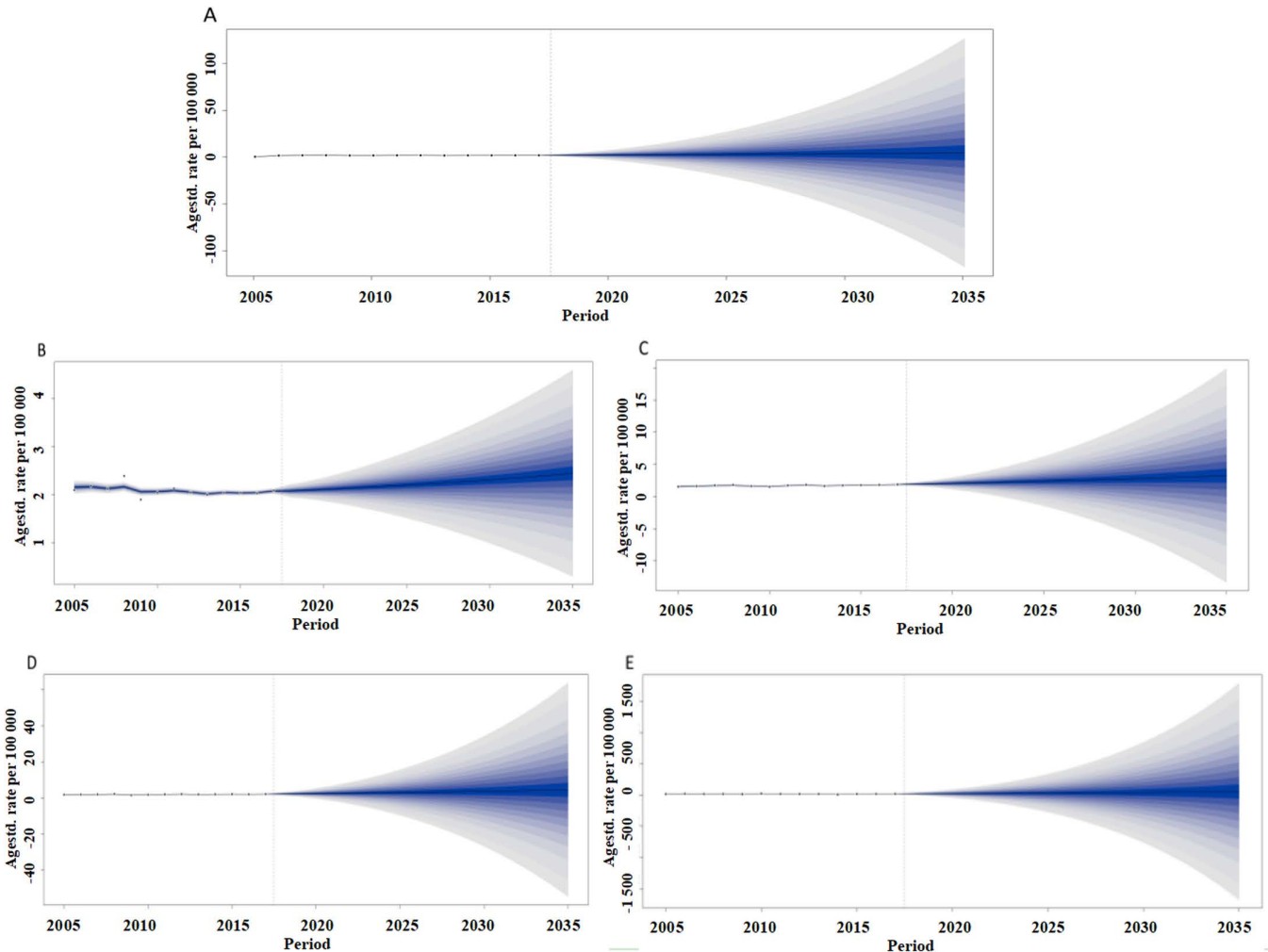

**Fig 3. Trends in the ASIR from 2005 to 2035 of NMSC in China.** (A)ASIR; (B) Men; (C) Women; (D) Urban; (E) Rural. NMSC, non-melanoma skin cancer.

significantly from 60.74 in 2018 to 37.3 in 2020, a significant downward trend that indicated that despite its rapid economic growth, China was also facing increasing environmental problems [31]. It may indirectly increase the health risks of NMSC through increased UV radiation, exposure to chemical pollutants and impacts on the immune system [21]. The increase in the pollution index demonstrated its association with increased incidence of NMSC. However, the incidence of NMSC did not decrease as the pollution index decreased, suggesting that there are additional factors that can influence the incidence of NMSC. With economic development and improved medical care, increased rates of early screening and diagnosis may also lead to an increase in the number of reported cases of NMSC.

The cohort effect peaked in 1930-1934 and then gradually declined, followed by a slight increase in 1995-1999. This suggests that individuals in different birth cohorts are significantly affected by external environmental factors and lifestyle changes. The high risk of the 1930-1934 birth cohort may be due to the high age of this birth cohort, and the accumulation of UV light is strongly associated with the incidence of NMSC. The slight increase in the risk of NMSC in the

1995-1999 birth cohort may be related to increased enhanced UV radiation and lifestyle changes in recent years. This suggests that targeted interventions should be taken in combination with the characteristics of different cohorts when formulating NMSC prevention measures.

The results of NMSC incidence prediction from 2019 to 2035 by BAPC model showed that the incidence of NMSC in China will continue to increase in the future, especially in rural and women. This is similar to global skin cancer prediction, which are likely to be the comprehensive result of future demographic changes, increased high-risk behaviors (e.g., outdoor recreation), and socioeconomic development. The continuing trend of population aging in China has also led to an increase in the number of people over 60 years old, so the risk of skin cancer will continue to rise. The incidence of NMSC is predicted to grow faster in rural areas, which may be related to lower awareness of sun protection and more outdoor activities [32], and only less than 20% of them often or always take sunscreen measures outdoors, which is also at a lower level compared with similar studies in other countries [33–34], and even lower than the results of relevant studies on urban populations in China [35]. It can be seen that residents in rural areas in China do not pay enough attention to daylight protection, and need to continue to publicize and educate.

This study has some limitations. The data was derived from the Annual Report of China Cancer Registry, and the original data was obtained from the national cancer registry rather than from a random sample, and thus the representativeness and extrapolation of the results to the entire population were inadequate. This study also has limitations in terms of timeliness, as there is generally a 3-year time delay in the latest cancer registry data. Due to insufficient available data, this study did not conduct a typing study of NMSC. At the same time, due to the lack of investigation and data on environmental influencing factors such as UV exposure, this study did not carry out an analysis of influencing factors, which will be carried out in future work. In addition, the observation period of this study is not long enough, which may lead to the instability of the prediction results. In the future, it is necessary to expand the sample for a longer period to obtain more stable results.

## 5. Conclusion

During 2005–2018, the incidence of NMSC in China showed a continuous upward trend nationwide, and will continue to rise by 2035. The ASIR of NMSC in men and urban was higher than that in women and rural, but the growth rate was reverse. It is particularly important to develop effective preventive measures and strategies. Reducing UV exposure is the main way to avoid NMSC. In addition, reducing environmental pollution and raising awareness of skin cancer are also core elements of strategy development.

## Supporting information

**S1 File. Supporting Information.**
(XLSX)

## Acknowledgments

We thank the National Cancer Center of China for their time and effort in preparing these publicly available data.

## Author contributions

**Conceptualization:** Su Liang, Juan Mei Cao, Xue Wang.

**Formal analysis:** Su Liang, Tao Sun, Xue Song Jia, Juan Mei Cao, Xue Wang.

**Funding acquisition:** Juan Mei Cao, Xue Wang.

**Methodology:** Su Liang, Tao Sun, Xue Song Jia, Juan Mei Cao, Xue Wang.

**Project administration:** Juan Mei Cao, Xue Wang.

**Software:** Su Liang, Tao Sun, Xue Song Jia, Juan Mei Cao, Xue Wang.

**Supervision:** Juan Mei Cao, Xue Wang.

**Writing – original draft:** Su Liang, Juan Mei Cao, Xue Wang.

**Writing – review & editing:** Su Liang, Juan Mei Cao, Xue Wang.

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
