## [Decision Letter · Decision Letter 0]

25 Nov 2024

PONE-D-24-40899Characterization and prediction of non-melanoma skin cancer incidence in China: Joinpoint regression and age-period-cohort modelPLOS ONE

Dear Dr. Cao, 

Thank you for submitting your manuscript to PLOS ONE. After careful consideration, we feel that it has merit but does not fully meet PLOS ONE’s publication criteria as it currently stands. Therefore, we invite you to submit a revised version of the manuscript that addresses the points raised during the review process. 

Please submit your revised manuscript by Jan 09 2025 11:59PM. If you will need more time than this to complete your revisions, please reply to this message or contact the journal office at plosone@plos.org . Please include the following items when submitting your revised manuscript:

We look forward to receiving your revised manuscript.

Kind regards,

Jobst Augustin, Associate Professor/Senior lecturer

Academic Editor

PLOS ONE

Journal Requirements:

2. Please note that funding information should not appear in the Acknowledgments section or other areas of your manuscript. We will only publish funding information present in the Funding Statement section of the online submission form. Please remove any funding-related text from the manuscript. 

3. Please note that your Data Availability Statement is currently missing the repository name or a direct link to access each database. If your manuscript is accepted for publication, you will be asked to provide these details on a very short timeline. We therefore suggest that you provide this information now, though we will not hold up the peer review process if you are unable.

**Additional Editor Comments:**

In principle, the submitted manuscript is an interesting topic, although the manuscript still needs to be thoroughly revised in some parts (see in particular the reviewers' comments). In addition, the following comments are made:

- Age is primarily considered in the analyses, although the introduction points out that other factors are also important. This should be better explained, especially as UV radiation is of particular relevance. The latter should be better analysed in terms of exposure (e.g. field work in the countryside) and explanation of urban-rural differences. The importance of the social situation should also be included.

- the discussion should be better structured, parts (e.g. lines 208-225) belong in my opinion also in the introduction

- The literature from some countries shows that leisure and holiday behaviour plays an important role in UV exposure (and skin cancer). How does this apply to China?

- On the one hand, the air pollution mentioned may also have a UV-reducing effect. How can this be assessed?

Reviewers' comments:

Reviewer's Responses to Questions

**Comments to the Author**

1. Is the manuscript technically sound, and do the data support the conclusions?

Reviewer #1: Yes

Reviewer #2: Yes

2. Has the statistical analysis been performed appropriately and rigorously? 

Reviewer #1: I Don't Know

Reviewer #2: No

3. Have the authors made all data underlying the findings in their manuscript fully available?

Reviewer #1: Yes

Reviewer #2: Yes

4. Is the manuscript presented in an intelligible fashion and written in standard English?

Reviewer #1: Yes

Reviewer #2: Yes

5. Review Comments to the Author

Reviewer #1: The subject is of great importance given the high prevalence of NMSC worldwide, particularly in a country with a large population such as China. The study makes a valuable contribution to the field of knowledge. The use of multiple statistical models ensures the robustness of the analysis and provides a comprehensive examination of incidence patterns. The use of data from the Annual Report of the China Cancer Registry provides a robust foundation for the study, enhancing the credibility of the findings.

However, the paper addresses age, gender, and regional influences on NMSC incidence but lacks a detailed examination of external factors such as UV exposure and environmental pollution, which could significantly affect the results. The presentation of results could be enhanced to improve clarity, for example through the use of tables and figures. It would be beneficial to present the key findings in a more accessible format for easier interpretation. The paper does not sufficiently address the study's limitations, particularly in relation to data representativeness and potential biases arising from the use of a national dataset rather than a random sample.

Overall, the paper provides valuable insights into NMSC incidence trends in China and employs suitable statistical methods for analysis. However, there is room for improvement regarding the depth of analysis, especially concerning external risk factors and public health implications. A clearer presentation of data and acknowledgement of limitations would enhance the paper's quality. I recommend revisions to address these points (if possible) before submission for publication.

Reviewer #2: This paper focuses on the incidence of NMSC and its percentage change (APC and AAPC) and trends in China. The incidences by gender and urbanity are shown. In addition, age-period-cohort models and predictions using BAPC were calculated. Differences were found between age, gender and region.

The paper is clear written and discusses the findings and probably causes in detail. It should be emphasized that different methodological approaches were used to describe and assess the incidence of NMSC.

Nonetheless, there are a few things to note, particularly concerning the figures and tables as well as some methodological aspects. Specifically:

- Table 1 is not easy to read, I would recommend using landscape format

- Fig.1: The lines and dots could be shown more clearly, for example a different choice of color, thicker lines or adjustment of the axes. The legend could also be larger.

- Fig. 2/Fig. 3: The axis labels are not legible.

- L.87: What does this formula refer to? A short description would be helpful.

- L.121: What does "I" stands for? What does the precision parameter do and how did you select it?

- L.139: 248/100,000 or 2.48/100,000? Please also check the other rates.

- L.187 95%-CI are given in the methods section. How these were calculated was not described. So I was wondering how an ASIR can be negative (as in the 95%-CI).

In addition, there are some minor points that I would like to mention:

- L.69: Can you please check the ICD10 classification for NMSC? C91-95 is coded for leukemia and D45-47 for polycythaemia vera.

- L.82: fited -> fitted

- L.159: The word “Tables” -> “Tables 1 and 2” or “Table 1, Table 2”

6. PLOS authors have the option to publish the peer review history of their article (what does this mean? ). If published, this will include your full peer review and any attached files.

**Do you want your identity to be public for this peer review?** For information about this choice, including consent withdrawal, please see our Privacy Policy .

Reviewer #1: No

Reviewer #2: No

---

## [Author Response · Author response to Decision Letter 1]

5 Dec 2024

PLOS ONE

Nov 4, 2024

PONE-D-24-40899

Characterization and prediction of non-melanoma skin cancer incidence in China: Joinpoint regression and age-period-cohort model

Dear editor and reviewers,

We appreciate editor and reviewers very much for their positive and constructive comments and suggestions on our manuscript entitled “Characterization and prediction of non-melanoma skin cancer incidence in China: Joinpoint regression and age-period-cohort model” (Manuscript ID: PONE-D-24-40899). Those comments are all valuable and very helpful for revising and improving our paper, as well as the important guiding significance to our researches. Based on these comments and suggestions, we have made careful modification on the original manuscript.

On the separate pages, we provided our response to the comments and suggestions, point by point, and highlighted the changes in the marked copy of the revision. We hope that our revision will be approved by the experts and reviewed favorably.

Sincerely,

Juan Mei Cao, MD

Additional Editor Comments:

In principle, the submitted manuscript is an interesting topic, although the manuscript still needs to be thoroughly revised in some parts (see in particular the reviewers' comments). In addition, the following comments are made:

- Age is primarily considered in the analyses, although the introduction points out that other factors are also important. This should be better explained, especially as UV radiation is of particular relevance. The latter should be better analysed in terms of exposure (e.g. field work in the countryside) and explanation of urban-rural differences. The importance of the social situation should also be included.

Response�;Thank you very much for your suggestions and questions. As for other influencing factors, such as ultraviolet exposure, due to the inavailability of data, the data used in this study came from cancer registration work, only epidemiological data, and no investigation on environmental factors has been carried out for the time being, and the limitations of the article were supplemented.

- the discussion should be better structured, parts (e.g. lines 208-225) belong in my opinion also in the introduction

- The literature from some countries shows that leisure and holiday behaviour plays an important role in UV exposure (and skin cancer). How does this apply to China?

- On the one hand, the air pollution mentioned may also have a UV-reducing effect. How can this be assessed?

Response�;Thank you very much for your suggestions. We have added to the introduction and deleted, modified and improved the discussion section. As for the relationship between environmental pollution and UV exposure, we further checked the data and found that the relationship between the two is more complicated, and different environmental pollutants have different effects on UV. Therefore, pruning the discussion section makes the discussion more reliable.

Reviewer #1: The subject is of great importance given the high prevalence of NMSC worldwide, particularly in a country with a large population such as China. The study makes a valuable contribution to the field of knowledge. The use of multiple statistical models ensures the robustness of the analysis and provides a comprehensive examination of incidence patterns. The use of data from the Annual Report of the China Cancer Registry provides a robust foundation for the study, enhancing the credibility of the findings.

1. However, the paper addresses age, gender, and regional influences on NMSC incidence but lacks a detailed examination of external factors such as UV exposure and environmental pollution, which could significantly affect the results.

Response�;Thank you very much for your advice. However, the purpose of this study is mainly to analyze the epidemic characteristics of non-melanoma skin cancer, and the influencing factors have not been analyzed for the time being. At the same time, since the data mainly come from cancer registration, only epidemiological data are available, and the investigation and analysis of influencing factors have not been carried out for the time being. Your suggestions are very important and significant, and we will add them to our future work.

In addition, we added the limitations of this study in lines 307-310 : “At the same time, due to the lack of investigation and data on environmental influencing factors such as UV exposure, this study did not carry out an analysis of influencing factors, which will be carried out in future work.”

2.The presentation of results could be enhanced to improve clarity, for example through the use of tables and figures. It would be beneficial to present the key findings in a more accessible format for easier interpretation.

Response�;Thank you very much for your advice. We modified and optimized the figures and tables to show the results with clear figures and tables, including 3 tables and 3 figures.

3.The paper does not sufficiently address the study's limitations, particularly in relation to data representativeness and potential biases arising from the use of a national dataset rather than a random sample.

Response�;Thank you very much for your question. This is indeed a limitation of this study. Due to the lack of full coverage of cancer registration in China, and the plan of gradual and comprehensive coverage, there is no random selection, so the overall representation is insufficient. However, the only data that can be used to describe and represent the incidence of cancer in China is this cancer registration work. But there are limitations to any study, so we have included a description of those limitations in the paper in lines301-304: “This study has some limitations. The data was derived from the Annual Report of China Cancer Registry, and the original data was obtained from the national cancer registry rather than from a random sample, and thus the representativeness and extrapolation of the results to the entire population were inadequate.” We hope to receive your forgiveness and understanding.

Reviewer #2: This paper focuses on the incidence of NMSC and its percentage change (APC and AAPC) and trends in China. The incidences by gender and urbanity are shown. In addition, age-period-cohort models and predictions using BAPC were calculated. Differences were found between age, gender and region.

The paper is clear written and discusses the findings and probably causes in detail. It should be emphasized that different methodological approaches were used to describe and assess the incidence of NMSC.

Nonetheless, there are a few things to note, particularly concerning the figures and tables as well as some methodological aspects. Specifically:

- Table 1 is not easy to read, I would recommend using landscape format

Response�;Thanks for your suggestion. We corrected it.

- Fig.1: The lines and dots could be shown more clearly, for example a different choice of color, thicker lines or adjustment of the axes. The legend could also be larger.

Response�;Thank you very much for your careful review, it has been revised.

Fig.1 Joinpoint regression in the incidence of NMSC in China, 2005–2018. NMSC, non-melanoma skin cancer.

- Fig. 2/Fig. 3: The axis labels are not legible.

Response�;Thank you very much for your valuable advice, we have made the Fig more clear.

Fig. 2 Age-period-cohort model of NMSC incidence in China, 2005–2018. (A) Nation; (B) Men; (C) Women; (D) Urban; (E) Rural. NMSC, non-melanoma skin cancer.

Fig. 3 Trends in the ASIR from 2005 to 2035 of NMSC in China. ( A )ASIR; (B) Men; (C) Women; (D) Urban; (E) Rural. NMSC, non-melanoma skin cancer.

- L.87: What does this formula refer to? A short description would be helpful.

Response�;Thanks for your careful reading. We added in lines ?: “The formula of the Joinpoint regression model is: ”

- L.121: What does "I" stands for? What does the precision parameter do and how did you select it?

Response�;Thanks for your question. We are very sorry for the misunderstanding caused by the fact that we did not show all the formulas. However, after further consulting the principle and process of BAPC model, we found that there are many formulas and the process is more complicated, so the reference literature describing its principle and method in detail is quoted in this paper. We got the result by running the R package directly.

In general, the Bayesian analysis method provides a method to obtain the hypothesis probability, and uses the Bayesian formula to combine the sample information with the prior information of unknown parameters to obtain the posterior information, and then deduces the unknown parameters according to the posterior information.

- L.139: 248/100,000 or 2.48/100,000? Please also check the other rates.

Response�;Thank you very much for your correct reminder, we have revised all.

- L.187 95%-CI are given in the methods section. How these were calculated was not described. So I was wondering how an ASIR can be negative (as in the 95%-CI).

Response�;Thank you very much for your question. We have re-run BAPC model and verified that the result is correct and there are indeed negative values in the interval. In addition, we have consulted many relevant literatures and found that they also have negative values, which may be due to the insufficient performance of the model or the short period of this study, which is the shortcoming of the study. Meanwhile, we have also tried other models, such as ARIMA model, but the results are not stable. It may be that the cycle is not long enough, and the predict data is not stable. We hope we can get your understanding and forgiveness.

For this reason, we have explained in the limited part in lines 310-312: “In addition, the observation period of this study is not long enough, which may lead to the instability of the prediction results. In the future, it is necessary to expand the sample for a longer period to obtain more stable results.”

- L.69: Can you please check the ICD10 classification for NMSC? C91-95 is coded for leukemia and D45-47 for polycythaemia vera.

Response�;Thank you very much for your review, we have revised it: “NMSC was diagnosed according to the International Classification of Diseases, 10th Revision (C44).”

- L.82: fited -> fitted

Response�;Thank you very much. We are very sorry for this error, and the whole text has been re-checked.

- L.159: The word “Tables” -> “Tables 1 and 2” or “Table 1, Table 2”

Response�;Thank you very much for your careful reading, this is our mistake, has been corrected.

---

## [Editor Report · Decision Letter 1]

11 Dec 2024

PONE-D-24-40899R1Characterization and prediction of non-melanoma skin cancer incidence in China: Joinpoint regression and age-period-cohort modelPLOS ONE

Dear Dr. Cao,

Thank you for revising the manuscript! However, after reviewing the revision, I have noticed some minor details that I would like you to check and correct.

We look forward to receiving your revised manuscript.

Kind regards,

Jobst Augustin, Associate Professor/Senior lecturer

Academic Editor

PLOS ONE

Journal Requirements:

Additional Editor Comments:

Thanks for the revision! However, I have noticed a few small things:

- Line 50-53: In my opinion, this is not written correctly. In 'western countries', the increase in NMSCs is mainly due to beauty ideals and related changes in leisure time (use of sunbeds, outdoor activites) and holiday behaviour (e.g. beach holidays). I cannot assess this in detail for emerging countries, but increasing air pollution (at least with particulate matter) does indeed lead to a reduction in UV. There are (conversely) European studies that show an increase in UV due to air pollution control measures. This should be presented in a more nuanced way, with sources.

- Line 83: Why did you delete "statistical analysis"? The structure made sense, didn't it? As it is now ("statistical analysis" in line 126) it is not quite right, because I understand "statistical analysis" as a generic term under which "joinpoint regression model" etc. can be summarised.

- Figure 1: In the illustration, the lines/dots (and colours) are virtually indistinguishable. I would recommend changing them, e.g. by combining solid and dashed lines.

---

## [Author Response · Author response to Decision Letter 2]

15 Dec 2024

PLOS ONE

Dec 13, 2024

PONE-D-24-40899

Characterization and prediction of non-melanoma skin cancer incidence in China: Joinpoint regression and age-period-cohort model

Dear editor and reviewers,

We appreciate editor and reviewers very much for their positive and constructive comments and suggestions on our manuscript entitled “Characterization and prediction of non-melanoma skin cancer incidence in China: Joinpoint regression and age-period-cohort model” (Manuscript ID: PONE-D-24-40899). Those comments are all valuable and very helpful for revising and improving our paper, as well as the important guiding significance to our researches. Based on these comments and suggestions, we have made careful modification on the original manuscript.

On the separate pages, we provided our response to the comments and suggestions, point by point, and highlighted the changes in the marked copy of the revision. We hope that our revision will be approved by the experts and reviewed favorably.

Sincerely,

Juan Mei Cao, MD

Journal Requirements:

Response�;Thank you very much for reminding us. After checking, we did not find any retracted articles, but several articles were in Chinese, and they were quoted to explain the situation in China.

Additional Editor Comments:

Thanks for the revision! However, I have noticed a few small things:

- Line 50-53: In my opinion, this is not written correctly. In 'western countries', the increase in NMSCs is mainly due to beauty ideals and related changes in leisure time (use of sunbeds, outdoor activites) and holiday behaviour (e.g. beach holidays). I cannot assess this in detail for emerging countries, but increasing air pollution (at least with particulate matter) does indeed lead to a reduction in UV. There are (conversely) European studies that show an increase in UV due to air pollution control measures. This should be presented in a more nuanced way, with sources.

Response�;Thank you very much for your professional advice. The reasons for the increase in the incidence of NMSC are really complex and multiple, and the relationship with pollutants is also complex, and we have modified it in lines 51-54: “In recent years, the incidence of NMSC has been on the rise globally, possibly due to changes in lifestyle, increased ultraviolet (UV) exposure, and an aging population. For example, UV light is considered to be a carcinogen, which can induce and promote the occurrence and development of cancer.”

- Line 83: Why did you delete "statistical analysis"? The structure made sense, didn't it? As it is now ("statistical analysis" in line 126) it is not quite right, because I understand "statistical analysis" as a generic term under which "joinpoint regression model" etc. can be summarised.

Response�;Thank you very much for your advice. We agree with you very much. It has been revised.

- Figure 1: In the illustration, the lines/dots (and colours) are virtually indistinguishable. I would recommend changing them, e.g. by combining solid and dashed lines.

Response�;Thank you very much for your question. We have modified it again and divided it into five figures, which is the same as the other two figures’ structures. We hope to get your satisfaction.

---

## [Editor Report · Decision Letter 2]

4 Feb 2025

Characterization and prediction of non-melanoma skin cancer incidence in China: Joinpoint regression and age-period-cohort model

PONE-D-24-40899R2

Dear Dr. Cao,

We’re pleased to inform you that your manuscript has been judged scientifically suitable for publication and will be formally accepted for publication once it meets all outstanding technical requirements.

Kind regards,

Jobst Augustin, Associate Professor/Senior lecturer

Academic Editor

PLOS ONE
---

## [Editor Report · Acceptance letter]

PONE-D-24-40899R2

PLOS ONE

Dear Dr. Cao,

I'm pleased to inform you that your manuscript has been deemed suitable for publication in PLOS ONE. Congratulations! Your manuscript is now being handed over to our production team.

Kind regards,

on behalf of

Dr. Jobst Augustin

Academic Editor

PLOS ONE